# Self-Referenced Multifrequency Phase-Resolved Luminescence Spectroscopy

**DOI:** 10.3390/s20195482

**Published:** 2020-09-24

**Authors:** Angel de la Torre, Santiago Medina-Rodríguez, Jose C. Segura, Jorge F. Fernández-Sánchez

**Affiliations:** 1Department of Signal Theory, Networking and Communications, University of Granada, 18071 Granada, Spain; atv@ugr.es (A.d.l.T.); segura@ugr.es (J.C.S.); 2By Techdesign S.L., 28500 Arganda del Rey, Madrid, Spain; santiago.medina@by.com.es; 3Department of Analytical Chemistry, University of Granada, 18071 Granada, Spain

**Keywords:** chemical sensor, luminescence spectroscopy, multifrequency, oxygen sensing, frequency response, quadrature detection, self-referenced analysis

## Abstract

Phase-resolved luminescence chemical sensors provide the analyte determination based on the estimation of the luminescence lifetime. The lifetime is estimated from an analysis of the amplitudes and/or phases of the excitation and emission signals at one or several modulation frequencies. This requires recording both the excitation signal (used to modulate the light source) and the emission signal (obtained from an optical transducer illuminated by the luminescent sensing phase). The excitation signal is conventionally used as reference, in order to obtain the modulation factor (the ratio between the emission and the excitation amplitudes) and/or the phase shift (the difference between the emission and the excitation phases) at each modulation frequency, which are used to estimate the luminescence lifetime. In this manuscript, we propose a new method providing the luminescence lifetimes (based either on amplitudes or phases) using only the emission signal (i.e., omitting the excitation signal in the procedure). We demonstrate that the luminescence lifetime can be derived from the emission signal when it contains at least two harmonics, because in this case the amplitude and phase of one of the harmonics can be used as reference. We present the theoretical formulation as well as an example of application to an oxygen measuring system. The proposed self-referenced lifetime estimation provides two practical advantages for luminescence chemical sensors. On one hand, it simplifies the instrument architecture, since only one analog-to-digital converter (for the emission signal) is necessary. On the other hand, the self-referenced estimation of the lifetime improves the robustness against degradation of the sensing phase or variations in the optical coupling, which reduces the recalibration requirements when the lifetimes are based on amplitudes.

## 1. Introduction

Chemical sensing based on quenched luminescence has become increasingly popular in the last decades due to its versatility and accuracy at a reasonable cost [1,2,3,4,5,6]. Different schemes for measuring the luminescence, including emission intensity and luminescence lifetime measurements are reported in the literature [7,8,9,10,11,12,13,14,15,16]. In the recent years, luminescence lifetime measurements estimated in the frequency domain (also known as phase-resolved luminescence) [17,18] have experimented an interesting development, for a number of reasons. On one hand, the involved instrumentation is relatively simple: the excitation light can easily be modulated with LEDs, the photodetectors appropriate for acquiring the luminescent response are inexpensive, and a complete instrument can be implemented with a microcontroller (for managing the excitation and emission signals), a digital-to-analog converter for the excitation, the light source, the sensing phase, the photodetector, and two analog-to-digital converters (one for the excitation and the other for the luminescent emission) [6]. On the other hand, even though the signal processing methods involved in phase-resolved lifetime estimations are relatively sophisticated [19,20,21], they can easily be programmed in a cheap microcontroller and/or a microcomputer, and portable and inexpensive instruments are feasible [5,6,20].

Classically, the phase-resolved luminescence spectroscopy is performed by applying a sinusoidal modulation to the light source, and the lifetime is estimated from the modulation factor and/or the phase shift (i.e., the amplitude ratio and the phase difference of the sinusoidal component in the emission and excitation signals) [5,18,22,23,24,25]. A chemical sensor based on phase-resolved luminescence includes a sensing phase presenting a dependence of the luminescence lifetime with the analyte concentration. From the amplitudes and phases of the excitation and emission signals, the modulation factor and phase-shift are measured, the corresponding lifetimes are estimated, and from them (and with appropriate calibration), the analyte concentration can be determined.

More recently, methods and instruments for multifrequency phase-resolved spectroscopy have been proposed [5,19]. A number of advantages are derived from the multifrequency analysis. Firstly, the lifetimes can be simultaneously estimated at several frequencies (without increasing the instrumentation requirements), and the analyte concentrations estimated at the different frequencies can be combined in order to obtain a more accurate and robust analyte determination. Secondly, if the excitation is configured as a rectangular signal (i.e., an on-off modulation instead of a sinusoidal or a multi-sinusoidal modulation), the hardware for the illumination source can be significantly simplified. Finally, multifrequency analysis allows the estimation of the uncertainties associated to the lifetimes and analyte concentrations, which can be applied to optimally combine the concentrations estimated at each frequency [21,26]. All these ideas have contributed to the development of portable, accurate, and inexpensive chemical sensors [6].

Conventionally, the phase-resolved determination of the lifetime requires the digitization of both the excitation and the emission signals because the modulation factor and phase shift are the amplitude and phase of the emission signal referred to those of the excitation signal. In this work, we propose to estimate the lifetime using only the emission signal, i.e., ignoring the excitation signal. The estimation of the lifetime using only the emission signal is possible if the emission signal contains several harmonics, because the amplitude and phase of one of the harmonics can be used as reference. The use of amplitudes and phases from the different harmonics referenced to one of them provides a new self-referenced phase-resolved luminescence spectroscopy where the excitation signal is not required for the analyte determination. Thus, in the proposed procedure, instead of using the excitation signal as reference for obtaining the lifetime, we propose to use the first harmonic of the multifrequency emission signal as reference, and the rest of the harmonics for the estimation of the lifetime. The proposed self-referenced method presents two advantages. On one hand, just one analog-to-digital converter (only for the emission signal) is required, which simplifies the instrument architecture and reduces instrument costs (simpler and cheaper micro-controllers can be used, no calibration required for multichannel interleaved analog-to-digital converters, etc.). On the other hand, the phase-resolved lifetime estimations based on amplitudes can benefit from the self-referenced procedure: in conventional amplitude-based lifetime estimations, recalibration is required in order to deal with the degradation of the sensing phase (emission amplitudes decreases as the sensing phase degrades, and therefore a calibration is necessary for compensating the sensing phase degradation) or variations in the optical coupling (which introduce an unknown factor in the ratio between emission and excitation amplitudes). Self-referenced amplitude-based lifetime estimation is immune to those problems because they affect the ratio between emission and excitation amplitudes, but not the ratio between the amplitudes of the different harmonics in the emission signal.

The idea of self-referenced sensing is not new in luminescence spectroscopy. It is commonly applied in the context of time-resolved luminescence lifetime estimation [18] and for intensity luminescence (usually at more than one emission wavelengths), with applications for pH and temperature instruments, in addition to analyte determination [27,28,29,30,31,32,33,34,35]. However, to the best of our knowledge, there is no previous proposal of self-referenced luminescence spectroscopy in the context of phase-resolved luminescence; therefore, the method described in this manuscript would provide the advantages of self-referencing to the accurate and inexpensive chemical sensors based on phase-resolved luminescence.

In this manuscript, in Section 2, we summarize the conventional procedure for the estimation of the lifetime in phase-resolved luminescence (i.e., using the excitation signal as reference) and describe in detail the proposed self-referenced procedure (i.e., using the first harmonic of the emission signal as reference and the rest of harmonics for the lifetime estimation). We have applied both the conventional and the proposed procedures to an oxygen measuring instrument, described in Section 3. The experimental results provided by both methods are compared and discussed in Section 4. Finally, Section 5 summarizes the work and provides the main conclusions.

## 2. Self-Referenced Estimation of the Luminescence Lifetime

### 2.1. Response of a Monoexponential Luminescent System

The response of a monoexponential luminescent system is described with a first order differential equation [6,18]:(1)$$\frac{d\;x_{em}\left(t\right)}{d\;t}=a\;x_{exc}\left(t\right)-\frac{x_{em}\left(t\right)}{\tau }$$
where $x_{em}$ is the signal describing the luminescent response, $x_{exc}$ is the signal used as excitation, *a* is a coefficient providing the relation between the excitation and emission amplitudes, and τ is the luminescence lifetime. In quenched-luminescence chemical sensors, the response is quenched by the presence of the analyte to be determined, and the changes in the lifetime $\tau_{q}$ are described by the Stern–Volmer equation:(2)$$\tau_{q}=\frac{\tau_{0}}{1+kC}$$
where *C* is the quencher concentration, $\tau_{0}$ is the lifetime at null quencher concentration and *k* is the Stern–Volmer constant [18,36,37]. The frequency response of the luminescent system can be derived from the differential equation by transforming it to the frequency domain [38,39,40,41]:(3)$$j\omega X_{em}\left(j\omega \right)=a\;X_{exc}\left(j\omega \right)-\frac{1}{\tau_{q}}X_{em}\left(j\omega \right)$$
where *j* is the imaginary unity and ω is the angular frequency, expressed in radians per second. The frequency response is obtained from the previous equation:(4)$$H\left(j\omega \right)=\frac{X_{em}\left(j\omega \right)}{X_{exc}\left(j\omega \right)}=\frac{\tau_{q}a}{1+j\omega \tau_{q}}$$
which is usually rewritten as:(5)$$H\left(j\omega \right)=M_{0}\frac{\tau_{q}}{\tau_{0}}\frac{1}{1+j\omega \tau_{q}}$$
where $M_{0}$ is the modulation factor (the ratio between the emission and excitation amplitudes) at low frequency and null quencher concentration. The frequency response H(jω) is a complex number (with modulus and argument) describing, for each frequency ω, the relationship between the excitation and emission signals. The modulus of H(jω) (also known as modulation factor) provides the ratio between the amplitudes of the emission and excitation sinusoidal components of frequency ω, while its argument represents the phase shift between both sinusoidal components [40].

### 2.2. Conventional Estimation of the Lifetime

The frequency response is useful for describing the phase-resolved luminescence chemical sensors. If the excitation signal $x_{exc}\left(t\right)$ is periodical with a fundamental frequency $f_{0}$, it contains harmonics at the angular frequencies $n\;\omega_{0}$ (with $\omega_{0}=2\pi f_{0}$ and n=1,2,3,…), and the emission signal $x_{em}\left(t\right)$ also contains harmonics at the same frequencies. The amplitudes and phases can be measured for both signals at the harmonics (by applying either a FFT analysis or a quadrature detection to the excitation and emission signals [5,19,20]), and the luminescent system can be characterized by the modulation factor and phase shift measured at the different harmonics.

Let $A_{exc}\left(n\right)$, $A_{em}\left(n\right)$, $\phi_{exc}\left(n\right)$, and $\phi_{em}\left(n\right)$ be the measured amplitudes and phases of the excitation and emission signals for the *n*-th harmonic. The modulation factor and the phase shift for the *n*-th harmonic are obtained, respectively, as:(6)$$m\left(n\right)=\frac{A_{em}\left(n\right)}{A_{exc}\left(n\right)}\;\;\;\phi \left(n\right)=\phi_{em}\left(n\right)-\phi_{exc}\left(n\right)$$

According to the frequency response expected for the monoexponential luminescent system, the modulation factor and the phase shift at the *n*-th harmonic are the modulus and the argument of the frequency response at the corresponding frequency:(7)$$m\left(n\right)=|H\left(j\;n\;\omega_{0}\right)|=M_{0}\frac{\tau_{q}}{\tau_{0}}\frac{1}{\sqrt{1+{\left(n\;\omega_{0}\tau_{q}\right)}^{2}}}\;\;\;\phi \left(n\right)=-\arctan\left(n\;\omega_{0}\tau_{q}\right)$$
and the lifetime estimations based on the modulation factor and phase shift at the *n*-th harmonic derived from the monoexponential model are, respectively [6,19]:(8)$$\tau_{m}\left(n\right)=\tau_{0}\frac{m\left(n\right)/m_{0}\left(n\right)}{\sqrt{1+{\left(n\;\omega_{0}\tau_{0}\right)}^{2}\left(1-{\left[m\left(n\right)/m_{0}\left(n\right)\right]}^{2}\right)}}\;\;\;\tau_{\phi }\left(n\right)=\frac{-\tan(\phi (n))}{n\;\omega_{0}}$$
where $m_{0}\left(n\right)$ is the modulation factor at null quencher concentration and at the *n*-th harmonic (which is usually easier than $M_{0}$ to be measured).

It should be noted that the modulation factor and the phase shift, used for estimating the lifetime, require the amplitudes and phases of both the emission and the excitation signals. In other words, in the conventional estimation of the lifetime, the amplitude and phase of the emission signal are referred to those of the excitation signal at each harmonic.

### 2.3. Lifetime Derived from the Amplitudes Referenced to the First Harmonic

In this work, instead of using the amplitudes and phases of the excitation signal as reference, we propose a self-referenced paradigm where the amplitudes and phases of each harmonic in the emission signal are referenced to those of the first harmonic.

Taking into account the frequency response of the monoexponential luminescent system, the amplitude of the *n*-th harmonic relative to that of the first one is:(9)$$\frac{A_{em}\left(n\right)}{A_{em}\left(1\right)}=\frac{A_{exc}\left(n\right)}{A_{exc}\left(1\right)}\frac{m(n)}{m(1)}=\frac{A_{exc}\left(n\right)}{A_{exc}\left(1\right)}\frac{\sqrt{1+{\left(\omega_{0}\tau_{q}\right)}^{2}}}{\sqrt{1+{\left(n\;\omega_{0}\tau_{q}\right)}^{2}}}$$
and therefore (since the ratio $A_{exc}\left(n\right)/A_{exc}\left(1\right)$ depends only on the excitation waveform) the self-referenced amplitude of the emission signal can be used for estimating the lifetime $\tau_{q}$. If we define the normalized self-referenced amplitude M(n) for the *n*-th harmonic as: (10)$$M\left(n\right)\equiv \frac{A_{em}\left(n\right)/A_{em}\left(1\right)}{A_{exc}\left(n\right)/A_{exc}\left(1\right)}$$
then the Equation (Equation 9) provides the self-referenced amplitude-based estimation of the lifetime, for the *n*-th harmonic referenced to the first one:(11)$$M^{2}\left(n\right)=\frac{1+{\left(\omega_{0}\tau_{q}\right)}^{2}}{1+{\left(n\;\omega_{0}\tau_{q}\right)}^{2}}\;\Rightarrow \;\tau_{sr.a}\left(n\right)=\frac{1}{\omega_{0}}\sqrt{\frac{1-M^{2}\left(n\right)}{n^{2}\;M^{2}\left(n\right)-1}}$$

### 2.4. Lifetime Derived from the Delay Referenced to the First Harmonic

If the *n*-th harmonic of the emission or excitation signal $x_{e}\left(t\right)$ has an amplitude and phase $A_{e}\left(n\right)$ and $\phi_{e}\left(n\right)$, then the corresponding sinusoidal component is:(12)$$x_{n.e}\left(t\right)=A_{e}\left(n\right)\cos\left(n\omega_{0}t+\phi_{e}\left(n\right)\right)$$
where *e* corresponds either to the emission or excitation signal. This sinusoidal component presents local maxima at the time values:(13)$$t_{e}\left(n\right)=\frac{1}{n\;\omega_{0}}\left(2\pi p-\phi_{e}\left(n\right)\right)$$
where *p* is an arbitrary integer. The delay of the local maxima of the *n*-th harmonic with respect to those of the first harmonic is:(14)$$\Delta t_{e}\left(n\right)=t_{e}\left(n\right)-t_{e}\left(1\right)=\frac{1}{n\;\omega_{0}}\left(2\pi p-\phi_{e}\left(n\right)\right)-\frac{1}{\omega_{0}}\left(2\pi p^{\prime }-\phi_{e}\left(1\right)\right)$$
and therefore, for the excitation and emission components, the self-referenced delay of the *n*-th harmonic can be written as:(15)$$\Delta t_{exc}\left(n\right)=\frac{1}{n\;\omega_{0}}\left(2\pi p+n\phi_{exc}\left(1\right)-\phi_{exc}\left(n\right)\right)$$
(16)$$\Delta t_{em}\left(n\right)=\frac{1}{n\;\omega_{0}}\left(2\pi p+n\phi_{em}\left(1\right)-\phi_{em}\left(n\right)\right)$$
where the integer *p* can be selected in order to make the sums in the parentheses in the interval ±π (in order to select the local maximum of the *n*-th harmonic with minimum delay to that of the first harmonic). If both delays are subtracted, the resulting difference can be related to the phase-shift of the frequency response of the luminescent system at the 1-st and the *n*-th harmonic:(17)$$\Delta t_{em}\left(n\right)-\Delta t_{exc}\left(n\right)=\frac{1}{n\;\omega_{0}}\left(2\pi p+n\;\phi \left(1\right)-\phi \left(n\right)\right)$$
and in the case of a monoexponential luminescent system this difference of the delays is:(18)$$\Delta t_{em}\left(n\right)-\Delta t_{exc}\left(n\right)=\frac{1}{n\;\omega_{0}}\left(2\pi p+\arctan\left(n\;\omega_{0}\tau_{q}\right)-n\arctan\left(\omega_{0}\tau_{q}\right)\right)$$

If we define an auxiliary function $F_{n}\left(x\right)$ as:(19)$$F_{n}\left(x\right)\equiv \arctan\left(n\;x\right)-n\arctan\left(x\right)$$
the Equation (Equation 18) can be rewritten as:(20)$$\Delta t_{em}\left(n\right)-\Delta t_{exc}\left(n\right)=\frac{1}{n\;\omega_{0}}\left(2\pi p+F_{n}\left(\omega_{0}\tau_{q}\right)\right)$$
and since $F_{n}\left(x\right)$ is a monotonically decreasing function with values between 0 and −π(n−1)/2 for positive values of *x*, the inverse function exists, and therefore the previous equation provides the self-referenced delay-based estimation of the lifetime for the *n*-th harmonic referenced to the first one:(21)$$\tau_{sr.d}\left(n\right)=\frac{1}{\omega_{0}}\;F_{n}^{-1}\left[n\;\omega_{0}\left(\Delta t_{em}\left(n\right)-\Delta t_{exc}\left(n\right)\right)-2\pi p\right]$$
where the integer *p* should be selected in order to keep the argument of the inverse function $F_{n}^{-1}\left(\right)$ in the interval [0,−π(n−1)/2]. The inverse function $F_{n}^{-1}\left(\right)$ does not admit an analytical expression, but several numerical methods could be applied to obtain the self-referenced delay based lifetime (for example, a tabulation-based interpolation of the function $F_{n}\left(x\right)$ or solving the Equation (Equation 20) with the Newton’s method). The function $F_{n}\left(x\right)$ is represented in Appendix A for several values of *n*.

As in the case of the self-referenced amplitude-based lifetime estimation, there is a dependence with the excitation signal (particularly with the delay $\Delta t_{exc}\left(n\right)$), but similarly, this delay depends only on the excitation waveform.

### 2.5. Summary of the Self-Referenced Amplitude- and Delay-Based Lifetime Estimations

In order to perform the self-referenced lifetime estimation, the following steps are necessary: [*i*] previously in the measuring procedure, from the excitation waveform, the amplitude ratio $A_{exc}\left(n\right)/A_{exc}\left(1\right)$ and the delay $\Delta t_{exc}\left(n\right)$ should be determined for each harmonic; [*ii*] in order to acquire a measurement, the emission signal $x_{em}\left(t\right)$ is recorded; [*iii*] from the emission signal, the amplitudes $A_{em}\left(n\right)$ and phases $\phi_{em}\left(n\right)$ are measured for each harmonic; [*iv*] using Equations (Equation 10) and (Equation 11), the self-referenced amplitude-based lifetimes $\tau_{sr.a}\left(n\right)$ are estimated; [*v*] from Equation (Equation 16), the delays $\Delta t_{em}\left(n\right)$ are obtained for each harmonic of the emission signal; and finally, [*vi*] from Equation (Equation 21), the self-referenced delay-based lifetimes $\tau_{sr.d}\left(n\right)$ are estimated. Obviously, the self-referenced estimations can be obtained for all the harmonics except the first one (which is used as reference).

### 2.6. Self-Referenced Lifetime Determination in a Simulation

In order to illustrate the self-referenced determination of the lifetime, a monoexponential luminescent system has been simulated and analyzed. The luminescent system has been defined with $M_{0}=1$ (modulation factor at C=0 and ω=0), $\tau_{0}$ = 100 μs (lifetime at C=0) and a Stern–Volmer constant k=0.5 kPa${}^{-1}$. The luminescent system has been simulated for null concentration (C=0, $\tau_{q}$ = 100 μs) and at a quencher concentration of C=5 kPa ($\tau_{q}$ = 28.57 μs). In the simulation, the excitation signal was a periodic repetition of rectangular pulses with amplitude 1 (in arbitrary units), with a fundamental frequency of 1 kHz (fundamental period of 1 ms) and 10% duty cycle (100 μs of pulse duration). The beginning of the first pulse has been arbitrarily set to $t_{0}$ = 226 μs.

The response of the luminescent system has been estimated for both concentrations by solving the differential equation with the finite difference method. Figure 1 shows the excitation signal (rectangular pulses) and the response for both values of the quencher concentration. As can be observed, at null concentration the luminescence amplitude and the lifetime are greater than in the presence of the quencher.

The excitation and emission signals have been analyzed with the quadrature detection method (or I/Q method) [5,19,20], and the sinusoidal components have been estimated from the corresponding amplitudes and phases for harmonics 1 to 9. The left panels in Figure 2 show, for each signal (excitation and emission at C=0 kPa and at C=5 kPa), the corresponding signal, the sinusoidal components corresponding to the different harmonics, and the composition of the different harmonics up to the 9-th one. The right panels of the figure show a detail of the harmonics for each signal.

As can be observed, for the excitation signal the harmonics are predictable: those corresponding to the rectangular signal of 10% duty cycle, with amplitudes $A_{exc}\left(n\right)=sinc\left(n/10\right)/5=\left\{0.1967,0.1871,0.1717,0.1514,\ldots \right\}$ for n=1,2,3,4,… (where sinc(x)=sin(πx)/(πx)), and null delays with respect to the first harmonic ($\Delta t_{exc}\left(n\right)=0$ for n=2,3,4,…).

For the emission signals, the delays referenced to the first harmonic are negative, as expected, and significant changes in the amplitudes and delays (referenced to the first harmonic) are observed when the analyte concentration increases, i.e., when comparing the figures in the center (for C=0 kPa) and the bottom (for C=5 kPa), which implies that the self-referenced amplitudes and delays can be applied (with an appropriate calibration) for the analyte determination.

From the amplitudes and phases measured for the emission signals at each harmonic, and the expected amplitudes and delays of the excitation signal ($A_{exc}\left(n\right)=sinc\left(n/10\right)/5$, $\Delta t_{exc}\left(n\right)=\;0$), the normalized amplitudes and delays referenced to the first harmonic have been estimated at each harmonic. Figure 3 represents the normalized amplitudes and delays referenced to the first harmonic (top panels) and the corresponding lifetimes (bottom panels), for the monoexponential system simulated at concentrations between 0 and 12 kPa. The self-referenced lifetimes were estimated at each harmonic from the normalized self-referenced amplitudes ($\tau_{sr.a}\left(n\right)$ using the Equation (Equation 11)) and the self-referenced delay ($\tau_{sr.d}\left(n\right)$ using the Equation (Equation 21)). As expected, at each concentration the estimated lifetimes (derived from the self-referenced amplitudes or delays and for the different harmonics) are identical, up to the precision provided by the time-increment in the finite difference method (applied to solve the differential equation in the simulation). In particular, the estimated lifetimes are 100 μs at C=0 kPa and 28.57 μs at C=5 kPa. Appendix A includes figures providing additional details of this simulation.

### 2.7. Determination of the Analyte Concentration

The proposed method provides two estimations of the lifetime (based on the self-referenced normalized amplitude and delay, respectively) for each harmonic (except for the first one, since it is used as reference): $\tau_{sr.a}\left(n\right)$ and $\tau_{sr.d}\left(n\right)$ for n=2,3,4,…. These estimations of the lifetime should be identical in the case of a monoexponential luminescent system (and if the measurements are not affected by noise). However, the photoluminescent systems usually deviate from the first order monoexponential model [6,18,19]. In such a case, the lifetime estimations are not expected to take the same value and are referred to as “apparent lifetimes”. By applying a specific calibration curve for each apparent lifetime, an independent estimation of the concentration is obtained from each one. These concentration estimations are affected by an uncertainty (manifested as an standard deviation when several measurements are acquired in the same conditions), in general different for each estimation, due to the instrumental noise (mainly electrical noise in the optoelectronic transducer and the preamplifier) and the error propagation from the primary measurements (amplitudes and phases) to the concentration estimations. According to statistics, the optimal combination of several independent unbiased estimators is obtained as the average of the individual estimators if they are affected by the same uncertainty, or as a weighted average if the uncertainties affecting each estimator are different. Therefore, the different concentration estimations can be combined into one robust measurement by applying weights inversely proportional to the square of the respective uncertainties [19,21,26].

## 3. Experimental Design

The proposed method for analyte determination (based on self-referenced estimations of the normalized amplitudes and delays) has been compared with the method conventionally applied in phase-resolved photoluminescence (using estimations of the modulation factor and phase-shift referred to the excitation signal) in experiments involving a sensing phase designed for oxygen detection. This sensing phase is a conventional Platinum(II) 5,10,15,20-*meso*-tetrakis-(2,3,4,5,6-pentafluorophenyl)-porphyrin immobilized in polystyrene (PS/PtTFPP) designed for measuring gaseous oxygen ($pO_{2}$) in the range 0–20 kPa. The sensing film was coated at the end of an optical fiber and was illuminated with an ultraviolet LED excited with a periodical rectangular signal with fundamental frequency $f_{0}=1715$ Hz (period $T_{0}$ = 583.09 μs) and 10% duty cycle (pulse duration $T_{on}$ = 58.31 μs). The luminescent response was acquired with a photomultiplier tube and a low-noise preamplifier, and both the excitation and the emission signals were recorded with a digital oscilloscope. Figure 4 includes a schematic diagram of the experimental set up and some electron microscopy images of the optical fiber core with the PS/PtTFPP coating. The sensing film, the procedure for coating the optical fiber, the instruments used for the data acquisition, and the I/Q method applied for measuring the amplitudes and phases of the excitation and emission signals at each harmonic are extensively described in our previous work [19].

Measurements were acquired at 13 oxygen concentrations between 0.00 and 12.00 kPa. At each concentration, 250 measurements were acquired, each one with a duration of 1 s. Half of them (125 at each concentration, randomly selected) were used for calibration purposes, while the other half was used for evaluation. The amplitudes and phases were estimated for the first 7 harmonics of the excitation and emission signals. Therefore, in the conventional procedure, each measurement provides 7 values of the modulation factor m(n) and phase-shift ϕ(n) (one for each harmonic) from the amplitudes and phases of the excitation and emission signals, while in the proposed procedure, each measurement provides 6 values of the self-referenced amplitude $A_{em}\left(n\right)/A_{em}\left(1\right)$ and the self-referenced delay $\Delta t_{em}\left(n\right)$ (for the harmonics 2nd to 7th), using only the emission signal.

## 4. Results and Discussion

### 4.1. One-Site and Two-Sites Model Fitting the Experimental Data

Figure 5 represents the response of the sensing phase (the modulation factor and the phase-shift) at each harmonic and concentration. This response was obtained from the data in the calibration partition (125 measurements at each concentration). Appendix A includes tables with the corresponding means and standard deviations. A monoexponential model (or one-site model) [18,19,42,43,44]:(22)$$H\left(j\omega \right)=\frac{M_{0}}{1+k\;C+j\omega \tau_{0}}$$
has been fitted to this experimental data. The fitting procedure obtains the monoexponential model providing a global fitting of the data for the frequencies and concentrations involved in the experiments. The resulting model parameters ($M_{0}=2.2089$, $\tau_{0}=65.0889\;\mu$s, and k=0.2826 (kPa)${}^{-1}$) provide a reasonable approximation of the luminescent system (with a determination coefficient $R^{2}=0.991649$), as can be observed in Appendix A. Similarly, a two-sites model [18,19,41,44,45,46]:(23)$$H\left(j\omega \right)=\frac{M_{0,1}}{1+k_{1}\;C+j\omega \tau_{0,1}}+\frac{M_{0,2}}{1+k_{2}\;C+j\omega \tau_{0,2}}$$
has been fitted to the experimental data. The two-sites model (with model parameters $M_{0,1}=1.7709$ a.u., $\tau_{0,1}=56.4362\;\mu$s, $k_{1}=0.3138$ (kPa)${}^{-1}$, $M_{0,2}=0.5908$ a.u., $\tau_{0,2}=139.8189\;\mu$s, and $k_{2}=0.1763$ (kPa)${}^{-1}$) significantly improves the description of the luminescent system response ($R^{2}=0.999122$), as can be observed in Appendix A.

### 4.2. Simulations of the Experiments Using the One-Site Model

The fitted one-site model has been used to simulate the luminescent system excited with a periodic rectangular signal with a fundamental frequency and pulse duration equal to those used in the experiments. This simulation has been carried out for concentrations between 0 and 12 kPa. The results of this simulation (similar to those previously discussed in the description of the proposed procedure) are presented in Figure 6 (including the self-referenced normalized amplitudes and delays, and the corresponding lifetimes). At each concentration, the obtained lifetimes are identical for the different harmonics and for amplitude- or delay-based estimations, as expected in a monoexponential model (because the estimations are based in a simulation using a one-site model with a well defined lifetime at each oxygen concentration). Appendix A includes figures with additional details about the signals and their harmonic decomposition for concentrations of 0.5 and 10 kPa.

### 4.3. Simulations of the Experiments Using the Two-Sites Model

Similarly, the fitted two-sites model has also been used to simulate the luminescent system in the experimental conditions, also for concentrations between 0 and 12 kPa. Figure 7 represents the self-referenced normalized amplitudes and delays and the corresponding apparent lifetimes, at the different harmonics and concentrations, estimated from the simulations (some complementary results of this simulation with additional details about the signals and their harmonic decomposition for concentrations of 0.5 and 10 kPa are included in Appendix A). Even though the emission signal and the harmonics look similar to those for the simulation with the one-site model, the self-referenced normalized amplitudes and delays provide apparent lifetime estimations that change depending on the harmonic and the procedure applied to obtain them (whether they were obtained from the self-referenced amplitudes or from the self-referenced delays).

The estimated apparent lifetimes obtained in the simulation for the two-sites model suggest several comments. Firstly, in a real luminescent system not matching a monoexponential model, at a given concentration the apparent lifetime is not expected to be constant (it changes with the harmonic and for amplitude- or delay-based estimations). Secondly, since the monoexponential model is an acceptable approach to the luminescent system, the apparent lifetimes of the two-site model (as well as the self-referenced normalized amplitudes and delays) show a dependence with the concentration similar to that of the monoexponential model, with apparent lifetimes in the same range of those for the monoexponential system. Finally, the strong dependence of the apparent lifetimes with the analyte concentration suggests that the self-referenced method can be applied for analyte determination (the dependence of the apparent lifetimes with the concentration is analyzed with more detail in the last figures of Appendix A). In order to obtain an accurate determination, each lifetime ($\tau_{sr.a}\left(n\right)$ and $\tau_{sr.d}\left(n\right)$) should be independently calibrated, and the analyte determinations derived from each apparent lifetime should be combined according to their respective uncertainties [19,21,26]. This situation is also found in conventional phase-resolved photoluminescence spectroscopy (and is particularly relevant in multifrequency photoluminescence sensors) [6,19].

### 4.4. Experimental Lifetimes Estimated with the Conventional Method

In the conventional multifrequency phase-resolved spectroscopy, the modulation factor and the phase shift are obtained, for each harmonic, from the amplitudes and phases of the excitation and emission signals. Figure 8 represents the experimental modulation factor and phase-shift (top panels) and the corresponding conventional apparent lifetimes (bottom panels) at each harmonic and concentration, averaged from the calibration partition. These results are complemented in Appendix A, with figures representing the individual measurements and the averages, and tables including the means and standard deviations. The distribution of the individual measurements at the different concentrations show that these estimations can be used for the oxygen determination. Due to the error propagation, the measurements obtained for the harmonics 5, 6, and 7 are very affected by noise (the dispersion is large compared to the differences associated to the oxygen concentration) and the utility of these harmonics for oxygen determination is limited. However, for the harmonics 1, 2, and 3, the differences for adjacent concentrations are larger than the dispersion, which guarantees a utility of these measurements for oxygen determination. As can be observed, the apparent lifetimes are not constant for each oxygen concentration (they change when are estimated from modulation factor or phase shift, and also change with the harmonic from which they are estimated), as expected for a non monoexponential luminescent system. This suggests independent calibrations for each apparent lifetime.

### 4.5. Experimental Lifetimes Estimated with the Self-Referenced Procedure

Figure 9 shows the experimental self-referenced normalized amplitudes and delays (top panels) and the corresponding estimated apparent lifetimes (bottom panels) at each harmonic (from harmonic 2nd to 7th) and concentration (between 0 and 12 kPa), averaged for the calibration partition. These results are complemented in Appendix A (for the amplitudes), Appendix A (delays), and Appendix A (apparent lifetimes) of the Appendix A, with figures representing the individual measurements and the averages, and tables including the means and standard deviations at each concentration and harmonic. These sections in the Appendix A also include the experimental results for the excitation signal (in addition to those for the emission signal).

As expected, the self-referenced amplitudes for the excitation signal do not depend on the oxygen concentration, and the values observed for each harmonic are very close to the expected ones for the waveform of the excitation signal (sinc(n/10)/sinc(1/10)). The self-referenced amplitudes of the emission signal have been normalized using the expected values of those for the excitation signal (therefore, the self-referenced normalized amplitudes for the emission signal do not require measurements of the excitation signal). Similarly, the self-referenced delays for the excitation signal are very close to null delays (as expected from the excitation waveform), and therefore no correction or normalization is applied to the self-referenced delays of the emission signal (which again do not require measurements of the excitation signal).

The plots with the individual measurements of the self-referenced normalized amplitudes, delays, and the corresponding apparent lifetimes (in Appendix A), with differences associated to the concentrations larger than the dispersion associated to noise, suggest the utility of the self-referenced method for oxygen determination. As in the case of the conventional method, the self-referenced apparent lifetimes are not constant for each concentration (changes associated to the harmonic and the amplitude- or delay-based estimations are observed), suggesting again an independent calibration of each apparent lifetime.

### 4.6. Accuracy in the Conventional and Self-Referenced $pO_{2}$ Determination

In order to obtain calibration curves for the inference of the oxygen concentration from the apparent lifetimes, each individual lifetime (from modulation factor or from phase-shift in the case of the conventional method, from the amplitudes or delays in the case of the self-referenced method, and from each available harmonic) has been fitted with a Demas model [42,43,47,48]. Appendix A shows the calibration parameters for each apparent lifetime. The root mean square (RMS) errors observed at each concentration have also been measured, since they are used to estimate the uncertainty of the individual $pO_{2}$ estimations, necessary to provide the combined $pO_{2}$ determination (the individual determinations are weighted according to the squared inverse of the uncertainty) [6,19,21,26].

Appendix A compares the RMS error in the oxygen determination using the conventional and the self-referenced methods, for each individual apparent lifetime and for each concentration. The combined oxygen determinations are detailed in Appendix A. Figure 10 summarizes the accuracy in the oxygen determination provided by the conventional and the self-referenced methods and Table 1 compares the results of the conventional and the self-referenced methods for the combined oxygen determinations, including the RMS error and the relative RMS error at each concentration. For the conventional method, the combination of the determinations based on $\tau_{m}$ and $\tau_{\phi }$ is useful to provide a good accuracy in all the concentration range. However, in the conventional method (and for the noise level in the experimental data), the combination of more than two harmonics does not improve the accuracy (or even produces a slight degradation because the lifetimes associated to the last harmonics are too noisy). On the other hand, for the self-referenced method, the combination of the different individual determinations provide significant improvement for all the harmonics.

The results obtained with the self-referenced method are significantly worse than those obtained with the conventional method. In the combined oxygen determinations, the RMS errors of the self-referenced method are increased, on average, by a factor of 2.79 with respect to the conventional method. This degradation is associated to the error propagation of the standard error from the primary measurements (amplitudes and phases) to the oxygen determination [21]. While in the conventional method a noisy emission harmonic and a clean excitation harmonic are involved, in the self-referenced method two noisy harmonics from the emission signal are involved. This guarantees that the variances of the self-referenced oxygen determinations are, at least, two times larger than those in the conventional method. In spite of this degradation, the accuracy provided by the self-referenced method is appropriate for an oxygen sensor, with a RMS error of 0.0160 and 0.2202 kPa, at oxygen concentrations 0.5 and 12 kPa $pO_{2}$, respectively (for the combination including all the available apparent lifetimes) and a relative error below 2% in the presented experiments (for all the considered concentrations except for 0.5 kPa).

The degradation in the RMS error observed for the self-referenced method is accompanied by two instrumental advantages. On one hand, the instrumental design is simplified in a self-referenced photoluminescence sensor, since only one signal (the emission one) has to be digitized (the excitation signal is not utilized in the self-referenced method). On the other hand, the conventional method using modulation factor is strongly affected by the degradation of the sensing film or changes in the optical coupling, through the factor $M_{0}$ in Equation (Equation 7). Changes of the constant $M_{0}$ (associated, for example, to a photodegradation) would cause a significant bias in the conventional $pO_{2}$ determination, requiring a recalibration of the instrument. In contrast, a reduction of the constant $M_{0}$ produces a reduction of the signal-to-noise ratio in the emission signal but not a bias in the self-referenced based oxygen determination. This significantly reduces the recalibration requirements of the oxygen sensors based on the proposed self-referenced method. Even though the experimental results presented in this manuscript include oxygen concentrations in the range 0–12 kPa using rectangular signals as excitation, the proposed procedure can directly be applied to multifrequency phase-resolved photoluminescence instruments using other excitation periodical signals, acquiring the emission signal with other photodetectors, designed for a different concentration range or using different sensing phases designed for oxygen detection or for other analytes.

## 5. Conclusions

In this manuscript, we have proposed a new self-referenced method for phase resolved luminescence spectroscopy. Compared with the conventional method (where the amplitudes and phases of the emission signal are referenced to those of the excitation signal), in the proposed method the amplitudes and phases of the different harmonics in the emission signal are referenced to those of the first harmonic. In the manuscript, we have presented the mathematical formulation of the proposed method, including the procedure for estimating the lifetimes and its application for the analyte determination.

The lifetime estimation provided by the self-reference method is consistent for first order (or monoexponential) luminescent systems. As in the case of the conventional phase-resolved method, for more complicated systems the proposed method provides reasonable estimations of the apparent lifetimes. In such luminescent systems, the lifetime estimations are not constant for a given concentration (they change when estimated from the self-referenced amplitude or delay, and for the different harmonics). However, as in the case of the conventional method, the apparent lifetimes provided by the self-referenced method can be used for the analyte determination.

The proposed self-referenced method (as in the case of the conventional one) allows the combination of several individual analyte determinations into a more robust one. The combination is obtained by applying weights inversely proportional to the square of the uncertainty of each individual determination. In the experiment presented in this manuscript, the uncertainties were estimated from the results in the calibration partition (to preserve the clarity in the method presentation). However, an uncertainty estimation based on spectral analysis of the emission signal could also be applied [21,26].

The accuracy in the oxygen determination provided by the proposed method is worse than that of the conventional method (the RMS errors are around 3 times greater in the proposed self-referenced method). This accuracy degradation is associated to the noise affecting the emission signal (much more important than that affecting the excitation signal) and the fact that the self-referenced estimations involve two noisy measurements (from two noisy harmonics in the emission signal) while the conventional estimations involve just one noisy measurement (the harmonic in the excitation signal is significantly less affected by noise). In case of instrumental requirements, this degradation can easily be compensated by increasing the duration of the recorded emission signals (each individual measurement was obtained from 1 s of emission signal) or by statistical accumulation of several measurements.

The self-referenced method provides interesting advantages for a luminescence sensor. On one hand, since only the emission signal is necessary, just one analog-to-digital converter is required, which simplifies the instrument architecture and reduces instrumental costs. On the other hand, the self-referenced method is invariant to the parameter $M_{0}$ (the amplitude ratio between the emission and excitation signals at null quencher concentration and low frequency), and therefore self-referenced lifetime estimations are not affected by a change in the parameter $M_{0}$ (due to degradation of the sensing film, changes in the preamplifier gain, changes associated to the optical coupling, etc.). In particular, the effect of the sensing phase degradation is a reduction in the luminescent response and the subsequent decrease in the signal-to-noise ratio (which would increase the standard error associated to the analyte determination) but it does not cause a bias in the self-referenced lifetime estimations or the analyte determinations. This significantly reduces the factory calibration and maintenance recalibration requirements, which constitutes a relevant practical advantage. Similarly, the self-reference method is invariant to group delay of the emission signal with respect to the excitation signal (for example, the time delay associated to certain photodetectors, like photo-multiplier tubes), which again reduces the instrument calibration requirements. Additionally, the relaxation in the calibration requirements would allow the use of luminescent materials initially discarded because of their limited long-term stability, which would open new perspectives in the design of chemical sensors based on luminescence.

## Figures and Tables

**Figure 1 sensors-20-05482-f001:**
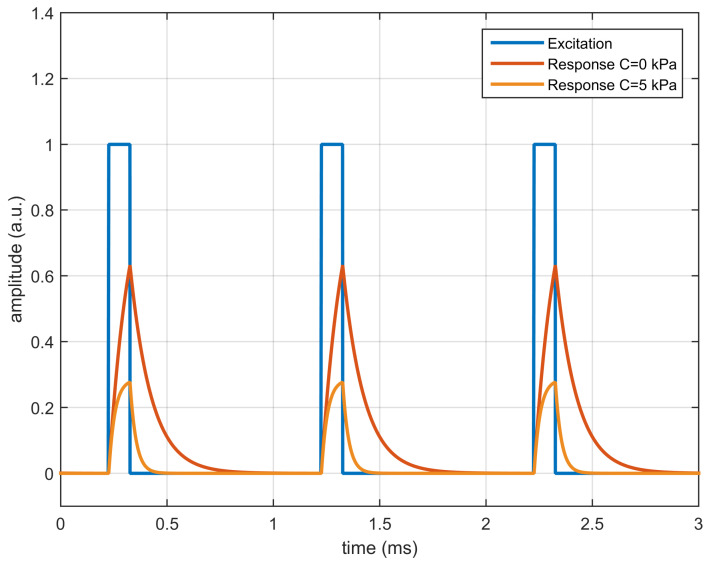
Simulated response of a first order luminescent system with $M_{0}=1$, $\tau_{0}$ =100 μs, k=0.5 (kPa)${}^{-1}$ excited with rectangular pulses of amplitude 1.0 a.u. presented at 1 kHz with 10% duty cycle, for quencher concentrations C=0 kPa and C=5 kPa.

**Figure 2 sensors-20-05482-f002:**
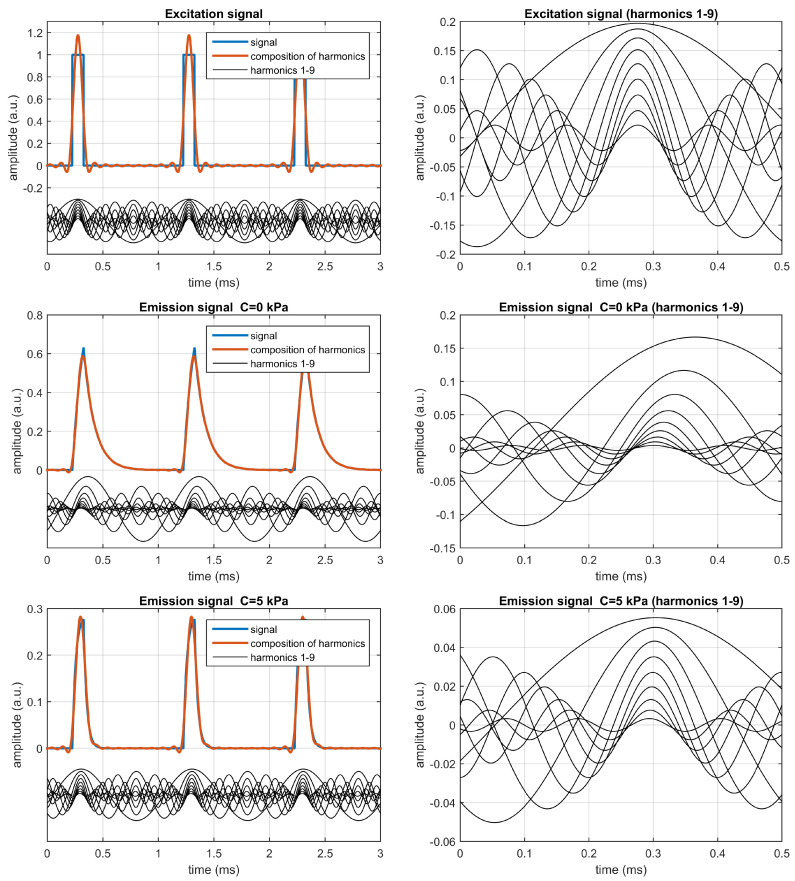
Analysis of the excitation signal (**top**) and the emission signals at C=0 kPa (**center**) and at C=5 kPa (**bottom**). The left panels represent the excitation or emission signals, the harmonics (from 1st to 9th), and the composition of the harmonics. The right panels show a detail of the harmonics for each signal. These plots correspond to the simulated monoexponential luminescent system with $M_{0}=1$, $\tau_{0}$ = 100 μs, and k=0.5 (kPa)${}^{-1}$, excited with rectangular pulses of amplitude 1.0 a.u. presented at 1 kHz with 10% duty cycle.

**Figure 3 sensors-20-05482-f003:**
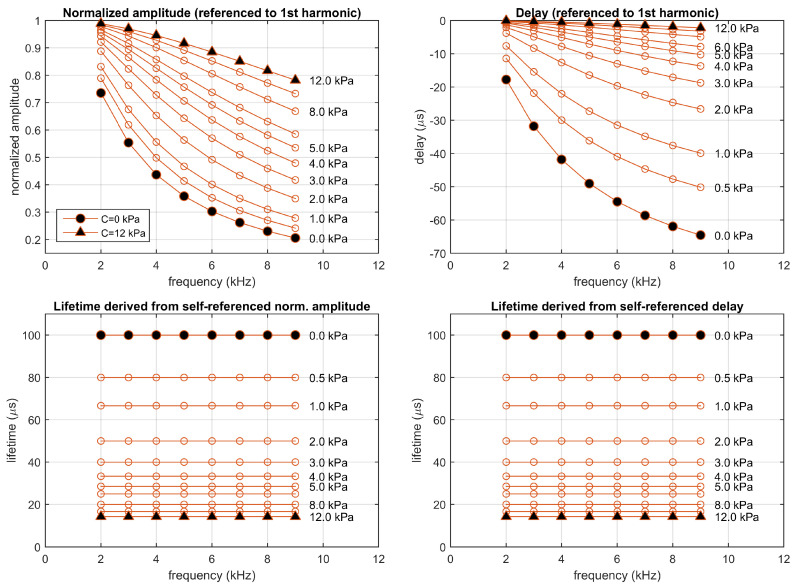
Simulation of a monoexponential luminescent system ($M_{0}=1$, $\tau_{0}$ = 100 μs, and k=0.5 (kPa)${}^{-1}$), excited with rectangular pulses of amplitude 1.0 a.u. presented at 1 kHz with 10% duty cycle. Top panels: Normalized amplitudes and delays referenced to the first harmonic; bottom panels: corresponding self-referenced lifetimes. Each point correspond to one harmonic (from 2nd to 9th). Simulations performed at concentrations 0, 0.5, 1, 2, 3, 4, 5, 6, 8, 10, and 12 kPa.

**Figure 4 sensors-20-05482-f004:**
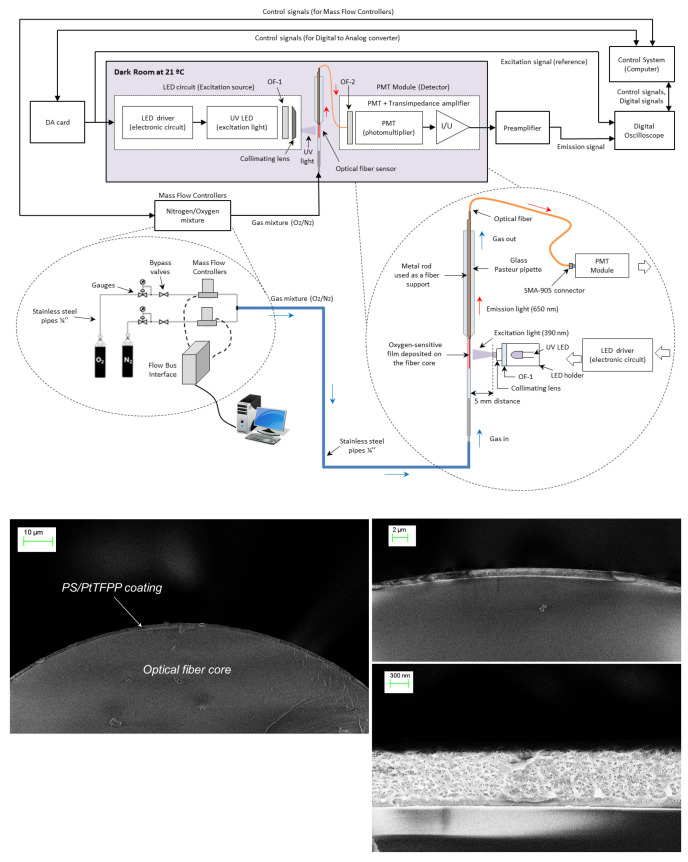
Experimental setup: (**top**) Schematic diagram of the experimental setup including the control system, the gas station, the excitation subsystem, the optical fiber sensor, and the acquisition subsystem; (**bottom**) scanning electron microscope (SEM) images of the optical fiber probe with the PS/PtTFPP coating.

**Figure 5 sensors-20-05482-f005:**
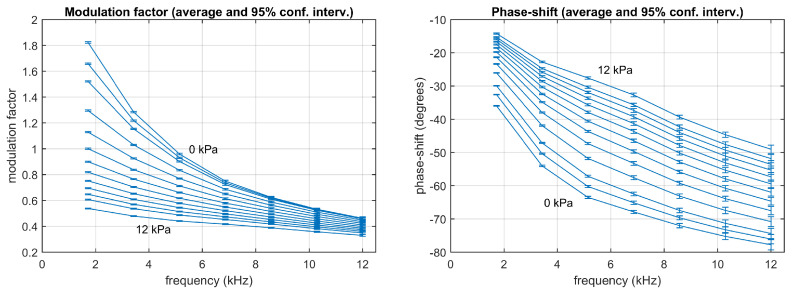
Experimental data: Modulation factor (**left**) and phase-shift (**right**) measured from the calibration partition for the harmonics 1st to 7th at the concentrations 0, 0.5, 1, 2, 3, 4, 5, 6, 7, 8, 9, 10, and 12 kPa. Each point represents the average using 125 measurements (the error bars represent the 95% confidence intervals).

**Figure 6 sensors-20-05482-f006:**
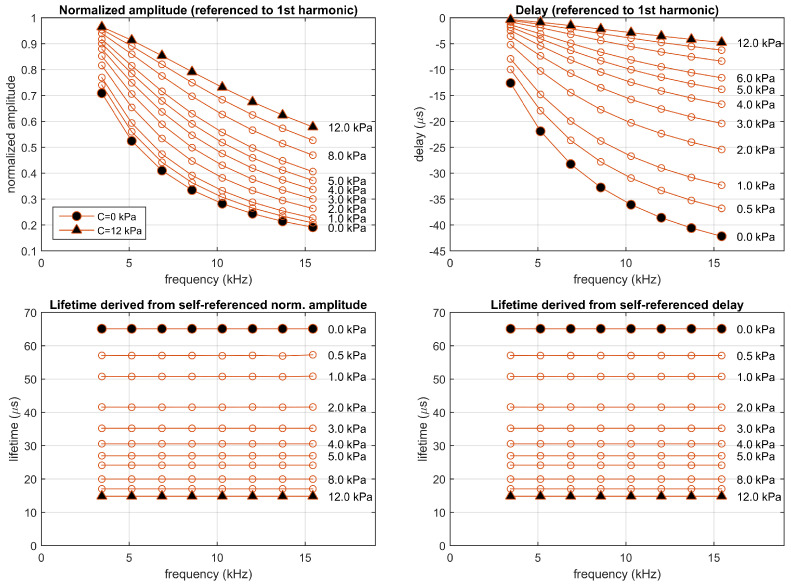
Simulation of the one-site model fitting the experimental luminescent system ($M_{0}=2.2089$, $\tau_{0}=65.0889\;\mu$s, and k=0.2826 (kPa)${}^{-1}$), excited in the experimental conditions (rectangular pulses presented at a fundamental frequency of 1715 Hz with a 10% duty cycle). **Top panels**: Normalized amplitudes and delays referenced to the first harmonic; **bottom panels**: corresponding self-referenced lifetimes. Each point correspond to one harmonic (from 2nd to 9th). Simulations performed at concentrations 0, 0.5, 1, 2, 3, 4, 5, 6, 8, 10, and 12 kPa.

**Figure 7 sensors-20-05482-f007:**
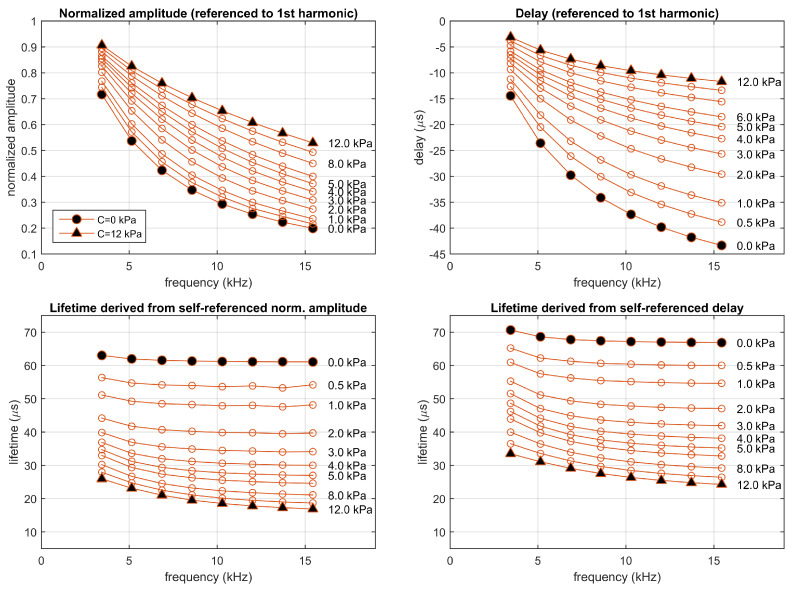
Simulation of the two-sites model fitting the experimental luminescent system excited in the experimental conditions (rectangular pulses presented at a fundamental frequency of 1715 Hz with a 10% duty cycle). **Top panels**: Normalized amplitudes and delays referenced to the first harmonic; **bottom panels**: corresponding self-referenced lifetimes. Each point correspond to one harmonic (from 2nd to 9th). Simulations performed at concentrations 0, 0.5, 1, 2, 3, 4, 5, 6, 8, 10, and 12 kPa.

**Figure 8 sensors-20-05482-f008:**
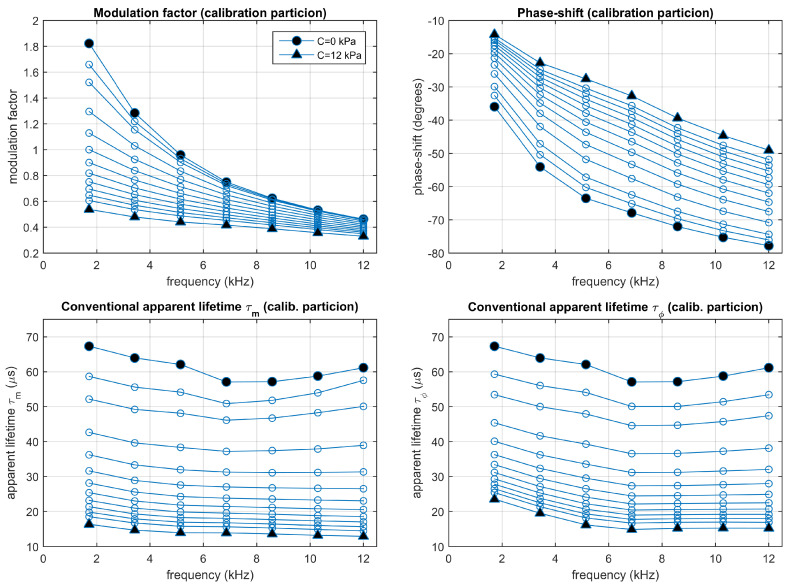
Experimental results: Conventional modulation factor and phase-shift (**top panels**) and the corresponding apparent lifetimes (**bottom panels**). Average values measured from the calibration partition. Each point corresponds to one harmonic (from 1st to 7th) and oxygen concentration (0, 0.5, 1, 2, 3, 4, 5, 6, 7, 8, 9, 10, and 12 kPa $pO_{2}$).

**Figure 9 sensors-20-05482-f009:**
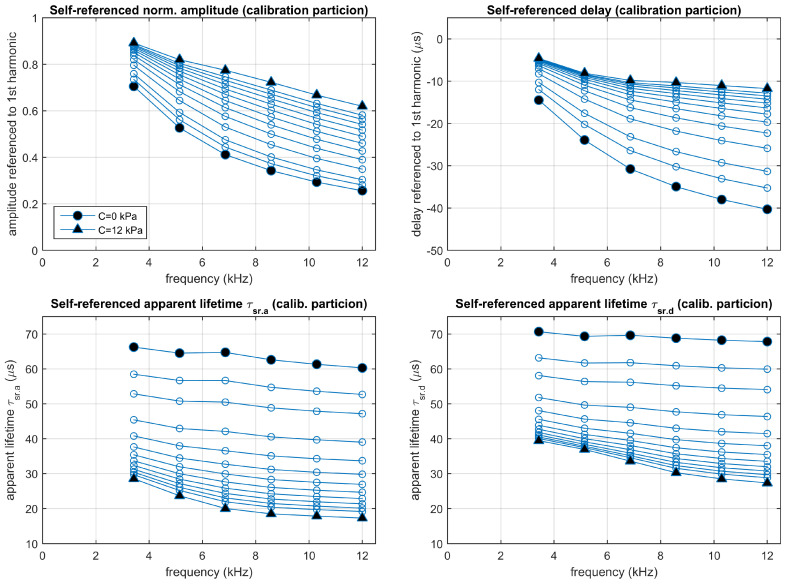
Experimental results: Self-referenced normalized amplitudes and delays, referred to the 1st harmonic (**top panels**), and the corresponding apparent lifetimes (**bottom panels**). Average values measured from the calibration partition. Each point corresponds to one harmonic (from 2nd to 7th) and oxygen concentration (0, 0.5, 1, 2, 3, 4, 5, 6, 7, 8, 9, 10, and 12 kPa $pO_{2}$).

**Figure 10 sensors-20-05482-f010:**
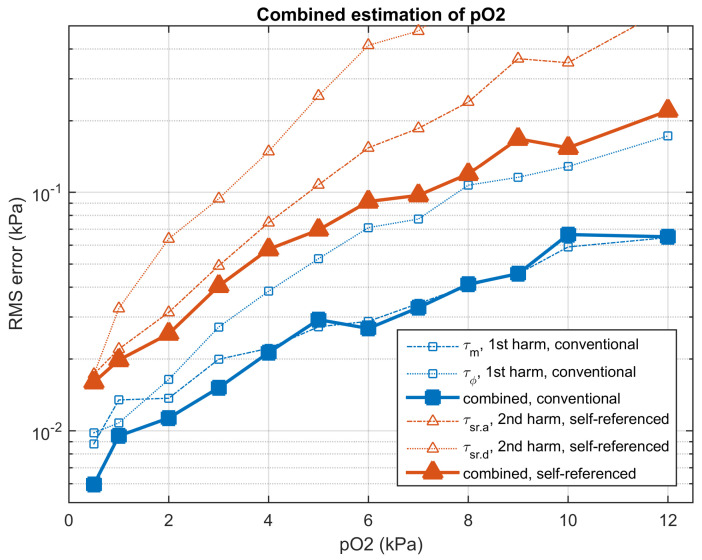
Root mean square (RMS) error (in kPa) in the oxygen determination for the conventional (blue squares) and self-referenced (red triangles) methods. The thin lines represent estimations using an individual apparent lifetime ($\tau_{m}$ or $\tau_{\phi }$ for the first harmonic in the conventional method, $\tau_{sr.a}$ and $\tau_{sr.d}$ for the second harmonic in the self-referenced method). The solid lines represent the oxygen determinations combining both apparent lifetimes for all the available harmonics. Experimental results obtained from the evaluation partition.

**Table 1 sensors-20-05482-t001:** Accuracy in the oxygen determination combining all the lifetimes from the different harmonics: comparison of the conventional and the self-referenced methods. Results corresponding to the evaluation partition.

RMS Error in ${\mathrm{pO}}_{2}$ Determination (kPa); in Parenthesis, Relative RMS Error (%)			
	**Conventional**	**Self-Referenced**	**Self-ref./conv.**
${\mathrm{pO}}_{2}$ **(kPa)**	**Harmonics 1–7**	**Harmonics 2–7**	**Ratio**
0.50	0.0060 kPa (1.20%)	0.0160 kPa (3.20%)	2.68
1.00	0.0095 kPa (0.95%)	0.0198 kPa (1.98%)	2.08
2.00	0.0113 kPa (0.56%)	0.0255 kPa (1.27%)	2.25
3.00	0.0152 kPa (0.51%)	0.0405 kPa (1.35%)	2.67
4.00	0.0213 kPa (0.53%)	0.0577 kPa (1.44%)	2.71
5.00	0.0292 kPa (0.58%)	0.0695 kPa (1.39%)	2.38
6.00	0.0269 kPa (0.45%)	0.0915 kPa (1.52%)	3.41
7.00	0.0328 kPa (0.47%)	0.0971 kPa (1.39%)	2.96
8.00	0.0412 kPa (0.51%)	0.1194 kPa (1.49%)	2.90
9.00	0.0456 kPa (0.51%)	0.1680 kPa (1.87%)	3.69
10.00	0.0665 kPa (0.66%)	0.1541 kPa (1.54%)	2.32
12.00	0.0651 kPa (0.54%)	0.2202 kPa (1.83%)	3.38
average	0.0309 kPa (0.62%)	0.0899 kPa (1.69%)	2.79

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
