# Peer review of "Self-Referenced Multifrequency Phase-Resolved Luminescence Spectroscopy"

_sensors, 2020, doi:10.3390/s20195482_

Round 1

Reviewer 1 Report

Authors propose a simplified self-referenced method of phase resolved luminescence spectroscopy that dies not require measurements of the excitation signal. This method is an interesting improvement for applications in portable instrumentation. To improve the manuscript, I would like to suggest the following: 

  • Details (results) of experiments and simulations are provided in Supplementary Materials, with the notes in the main text. It makes it not suitable for the reader to go back and forth from the paper to the supplementary files. It would be better to include the important results in the body of the paper. 
  • The Section 3 is very short. Authors cite their previous work for the readers to find out about the experimental design from another paper. It would be great to include at least a schematic / photos of the experimental system in this section. 
  • Section 2.4 and 2.5 can be combined.

In Section 2.6, only one quencher concentration was modeled and compared to the zero quencher concentration. I think that modeling of many quencher concentrations, especially close to each other, would benefit in determining the "resolution" of this method in concentration measurements. 

Line 228 - what is a numerical precision of simulations? 

Line 242 - authors mentioned 'respective uncertainties' - it would be great to explain shortly what these uncertainties are and their range. 

The proposed method allows simplifying the sensing system by elimination of a channel related to recording and processing of the excitation signal. Another advantage that the factor M0, often affected in measurements and needed to be checked, is not used. However, the noise (that authors mentioned) and the RMS error of the proposed method will increase. It would be great to provide analysis of such error increase in this self-referenced method (e.g., how it affects the 'resolution' in quencher concentration measurements) and compare to the conventional method (with measurements of the excitation signal). Thus, a 'trade-off' between the error increase and simplification of the measurement process would be determined. 

Reviewer 2 Report

In "Self-referenced multifrequency phase-resolved luminescence spectroscopy" authors investigated the possibility to apply self-reference calibration to estimate the luminescence apparent lifetime. The approach is initially validated utilizing simulated case studies and then applied to an experimental dataset where oxygen concentration is detected by utilizing a Pt-porphyrin immersed in a polymeric matrix. Utilizing the first harmonic as reference removes the need for measuring excitation contribute and for periodic recalibration. Along with these benefits, utilizing part of emission signal as reference drastically increases the noise of measurements mainly due to error propagation. Results show that in case of the aforementioned sensor, it can be apply with the proposed method in the 0-12 kPa range. In the reviewer opinion the manuscript is well written and conclusions are supported by both simulations and results. 

As minor consideration, in the reviewer opinion it would be helpful if authors include some applications where the oxygen sensor with reduced operating range (0-12kPa) can be applied. Finally in the manuscript some typos are present, please check them.

Round 2

Reviewer 1 Report

Authors addressed the reviewers's comments. I would like to suggest to accept the manuscript in the revised form.